# Gender Related Changes in Gene Expression Induced by Valproic Acid in A Mouse Model of Autism and the Correction by S-adenosyl Methionine. Does It Explain the Gender Differences in Autistic Like Behavior?

**DOI:** 10.3390/ijms20215278

**Published:** 2019-10-24

**Authors:** Liza Weinstein-Fudim, Zivanit Ergaz, Gadi Turgeman, Joseph Yanai, Moshe Szyf, Asher Ornoy

**Affiliations:** 1Department of Medical Neurobiology, Hebrew University Hadassah Medical School, Jerusalem 91120, Israel; liza.weinstein-f@mail.huji.ac.il (L.W.-F.); Zivanit@hadassah.org.il (Z.E.); josephy@ekmd.huji.ac.il (J.Y.); 2Department of Molecular Biology and Pre-Medical Studies, Ariel University, Ariel 40700, Israel; gadit@ariel.ac.il; 3Department of Pharmacology and Therapeutics, McGill University Medical School, Montreal, QC H4A3J1, Canada; moshe.szyf@gmail.com

**Keywords:** ASD, epigenetics, mice, postnatal VPA injection, SAM, gene expression, nanostring

## Abstract

In previous studies we produced autism like behavioral changes in mice by Valproic acid (VPA) with significant differences between genders. S-adenosine methionine (SAM) prevented the autism like behavior in both genders. The expression of 770 genes of pathways involved in neurophysiology and neuropathology was studied in the prefrontal cortex of 60 days old male and female mice using the NanoString nCounter. In females, VPA induced statistically significant changes in the expression of 146 genes; 71 genes were upregulated and 75 downregulated. In males, VPA changed the expression of only 19 genes, 16 were upregulated and 3 downregulated. Eight genes were similarly changed in both genders. When considering only the genes that were changed by at least 50%, VPA changed the expression of 15 genes in females and 3 in males. Only Nts was similarly downregulated in both genders. SAM normalized the expression of most changed genes in both genders. We presume that genes that are involved in autism like behavior in our model were similarly changed in both genders and corrected by SAM. The behavioral and other differences between genders may be related to genes that were differently affected by VPA in males and females and/or differently affected by SAM.

## 1. Introduction

Autism spectrum disorder (ASD) is a complex neurodevelopmental disorder characterized by impaired social communication and social interactions and restrictive stereotyped behaviors and interests. ASD core symptoms are frequently accompanied by developmental delay, anxiety and cognitive deficits [1,2,3]. Currently, the prevalence of ASD is 1 in 59 children, at an earlier onset of children prior to 3 years of age [4]. ASD is more prevalent in males, with the average male to female ratio 4:1 [4]. This male bias that can be partially explained by specific genetic differences between males and females had various impacts on both research and clinical practice [5]. Furthermore, autistic male and female demonstrate different phenotypes; females were usually reported to have lower social and communicative problems and fewer restricted and repetitive behaviors than males [6,7,8]. The age of diagnosis is also on average later in autistic females than males, and females may often be missed by current diagnostic procedures [9].

Although ASD is a highly heritable disorder, prenatal and early postnatal exposures to environmental toxicants (i.e., pollutants, insecticides and pesticides) or maternal infections as well as epigenetic alterations are hypothesized to contribute to the development of ASD. Nevertheless, the underlying cause is still unclear [1,2,3,10]. Moreover, currently there are no biochemical or molecular markers for the diagnosis of ASD.

Animal models enable the use of preclinical tools to understand the role of genetic mutations and environmental factors in the etiology of ASD. Hence, there are several rodent models in which ASD-like symptoms were produced by exposure of the pregnant dams to a teratogenic agent, especially valproic acid (VPA), during different stages of gestation, and a few following early postnatal insults [2,10,11].

VPA is an antiepileptic drug that has also been used for bipolar disorder, migraine headaches and as a mood stabilizer [12,13]. VPA exposure during pregnancy, especially during the organogenesis period, is associated with various teratogenic effects including high risk of major anomalies, mainly spina bifida. VPA during pregnancy also leads to 5–10 fold increased rate of ASD in the offspring [3,14]. Similarly, rodents exposed to VPA during different stages of pregnancy exhibit behavioral deficits including delayed developmental milestones, stereotypic and self-injurious behavior and impaired social behavior similar to those observed in autistic patients [11,15,16,17,18,19]. Therefore, VPA rodent model became a robust, widely used, environmental preclinical model of ASD with face construct and predictive validity [20].

Interestingly, while the male to female ratio observed in human ASD is 4:1, the male to female ratio in ASD children exposed to VPA during pregnancy is nearly 1:1 [21] or 2:1 [22].

As in human, there are behavioral differences between male and female rodents exposed to VPA. Male mice exposed to VPA exhibited social impairment and reduced social interaction, manifested by lack of preference for a stranger mouse in the three-chamber test [23,24,25]. On the other hand, increased repetitive behavior and anxiety-like behaviors was reported in both males and females [23,26].

It is important to remember that most cases of ASD in human are genetic in origin and not related to the use of specific drugs or chemicals during pregnancy. There are also distinct neurobehavioral differences in human with ASD when compared to animal models. For example, there is a wide range of cognitive abilities in human from normal to severe retardation (1,2), while learning is often impaired in the VPA animal model.

The mechanism(s) by which VPA exposure during pregnancy causes autistic-like behaviors in both human and rodent offspring are still far to be clear. Among the proposed mechanisms are: Folic acid deficiency, effects on Wnt signaling, increased oxidative stress [11], alternations in serotonin homeostasis [27] and in the activity of gamma amino butyrate (GABA) neurotransmitter, and neuronal spine density changes. Other suggested pathways are derangement of the serotonergic, dopaminergic and/or oxytocinergic systems. In addition, VPA is an epigenetic modulator and a potent histone deacetylase (HDAC) class l and ll inhibitor [9,11,12,13]. Histone deacetylases removes acetyl groups from the tail of core histones, and regulates chromatin condensation and gene expression. Mice exposed prenatally to VPA demonstrate transient increase in acetylated histone levels [28].

Recently, more and more studies suggest that multifactorial conditions of ASD result from epigenetic changes, i.e., heritable changes in gene expression via a number of mechanisms including DNA methylation, histone modifications and ATP-dependent chromatin remodeling, without changing the underlying DNA sequence.

Choi et al. [29] demonstrated that the effect of prenatal VPA exposure in offspring could be paternally transmitted from the first up to the third generation. They detected autistic-like behaviors and increased postsynaptic markers of excitatory neurons in VPA-exposed F2 and F3 generations similar to those impairments detected in F1 VPA prenatally exposed mice. Furthermore, because the transmission experiments were performed with non- exposed female mice, the transmission is paternal and it is not influenced by any in utero exposures. They also reported congenital malformations only in F1 generation of VPA exposed mice, suggesting that VPA-induced congenital defects are mediated by a mechanism other than transmissible epigenetic changes. Their findings support the transgenerational epigenetic inheritance in the etiology of ASD.

So far, genetic studies identified more than 1000 genes that contribute to ASD risk [30]. Nevertheless, specific cellular abnormalities that link genes with behavior remain obscure. Most studies exploring gene alternation in VPA exposed rodents focused on specific set of genes or genes involved in the process of interest [31,32,33]. Only a few studies demonstrated comprehensive genetic analysis. Zhang et al. [34] performed transcriptome analysis in the prefrontal cortex of male rats prenatally exposed to VPA. They reported 3228 differently expressed genes and 637 genes differently spliced in the VPA group compared to controls, including genes involved in neurological diseases such as Huntington’s disease, Alzheimer’s disease and Parkinson’s disease. VPA also changed the expression of genes associated with neurogenesis, generation of neurons, neuron projection development, neuron differentiation and synaptic development.

Kotajima-Murakami et al. [35] treated prenatally VPA exposed pups with intraperitoneal injection of rapamycin for 2 consecutive days. The mammalian target of rapamycin (mTOR) signaling pathway plays a crucial role in cell growth, proliferation and metabolism. Mice prenatally exposed to VPA exhibited the aberrant expression of genes associated with the mTOR signaling pathway, and rapamycin treatment recovered changes in the expression of some genes.

Hill et al. [36] explored changes in methylation of CpG islands in the frontal cortex of mice prenatally exposed to VPA as well as mice exposed to other toxicants including lead and manganese. They reported lower regulation and overexpression of Chd7 gene essential for neural crest cell migration and patterning in all treatment groups.

Several brain structures and functions have been suggested to underlie behavioral abnormalities of ASD including the prefrontal cerebral cortex [37], the cerebellum, the hippocampus and the basolateral amygdala [38]. The prefrontal cerebral cortex is affected in ASD at the neuronal development and synaptic functionality levels. Studies found alternations such as Hyper-Connectivity and Hyper-Plasticity in the pyramidal network in the medial prefrontal cortex [39]. Thus, we decided to focus on this part of the brain in our study.

S Adenosyl methionine (SAM) is the principal biological methyl donor involved in multiple biochemical reactions and critical for regulation of cell growth, differentiation, and function and biosynthesis of hormones and neurotransmitters [40]. SAM has also been shown to be involved in reduction of oxidative stress [41,42,43,44]. Villalobos et al. [41] showed that SAM modulates cellular oxidative status, mainly by inhibiting lipid peroxidation and enhancing glutathione system in the brain of rat model of brain ischemia-reperfusion.

Reduction in SAM levels has been found in neurological conditions such as Alzheimer disease [45]. Numerous studies have therefore explored the efficiency of treatment of depression and other neuropsychiatric conditions by SAM administration [46,47,48,49]. A few studies indicated that gender might impact the antidepressant efficacy of SAM, with greater therapeutic effect found in males [50].

In our recent study [23] we injected four-day-old mice with a single dose of 300 mg/kg of VPA and tested them on a variety of neurobehavioral tests during postnatal days 50 to 59. The mice exhibited typical ASD-like behavior, with differences between males and females in some of the tests. Lower preference for social novelty and stereotyped repetitive behaviors were more prominent in males, while impaired re-learning ability were more prominent in females. In addition we evaluated oxidative stress parameters in the prefrontal cortex and liver on day 60. Enhanced oxidative stress was observed in the prefrontal cortex, demonstrated by changes in the activity of Superoxide dismutase (SOD) and Catalase (CAT) enzymes and increased lipid peroxidation in VPA treated mice. In general oxidative stress parameters were more prominent in females. There were no changes in the redox potential of the liver, implying that the oxidative stress was induced only in the brain as a result of the epigenetic changes induced by the VPA. The co-administration of VPA and SAM alleviated most ASD like neurobehavioral symptoms and normalized the redox potential in the prefrontal cortex [23].

Our current research focuses on the gene expression changes in the VPA treated mice. In this study, we aimed to evaluate the effects of early postnatal VPA administration on ASD like behavior and the therapeutic effect of SAM on the VPA exposed mice by measuring the changes in the expression of genes involved in various neurobiological pathways. In addition we aimed to evaluate the differences in the gene expression between males and females in order to understand the cause of the ASD like behavioral variance between genders.

## 2. Results

As there were significant gender differences in the expression of genes, the data is described separately for males and for females. In females, VPA injection induced statistically significant changes in the expression of 146 genes, 71 genes were upregulated and 75 genes were downregulated. In males, VPA changed the expression of 19 genes, 16 were upregulated and 3 genes were downregulated. Nine of these genes were similarly changed in both genders (See Figure 1 and Figure 2).

Gene ontology functional enrichment analysis for the genes significantly changed by VPA in females revealed several pathways associated with Huntington disease, Alzheimer’s disease, Prostate cancer, focal adhesion, calcium and PIK3-signaling (Table 1). Of special importance for our study are pathways of associated diseases involving cognitive impairment, namely Alzheimer’s disease and Huntington’s disease (Appendix A). In both pathways, key genes were upregulated in VPA exposed females (i.e., APP & Htt). Furthermore, Huntington’s disease pathway share mechanisms with epigenetic modulation of DNA and gene repression, including HDAC - a putative target for VPA (Appendix A). However, in males, no statistical significant pathway was detected. We therefore, continued to explore the data by focusing on specific genes with a cut-off of 50% de-regulation of expression, and their possible role as key genes.

Figure 1A,B represent heat map of mRNA levels of genes involved in neuropathological pathways at the prefrontal cortex. For each gene the expression was normalized to the geometrical mean of 7 housekeeping genes and the negative and positive technical controls. Each vertical column represents one animal belonging to the treatment group indicated in the upper panel while each horizontal lane represents the normalized mRNA counts for one gene. The colors represent the expression of each gene among the different treated animals (red and blue represent strong and weak expression, respectively). As observed, there were many genes whose expression was significantly changed by VPA. More of these genes were up or downregulated in females compared to males.

For strict selection we decided to focus on genes with at least 50% change by VPA. With the restriction of 50% change in comparison to controls, there were 15 genes in females and 3 genes in males whose expression was changed by VPA. Only one of these genes, Nts, was similarly downregulated in both genders, as described in Table 2 and Table 3.

Table 2 shows the percent of change by VPA compared to controls for each gene in females, whether the change is corrected by SAM administration or not. Six genes were upregulated and 9 were downregulated. The affected genes are involved in various pathways; 6 are involved in neurotransmission, 5 in neuroplasticity and 6 in neuroinflammation. Note that 4 of the genes also appear in Sfari database for genes reported in human ASD. These genes are involved in neurotransmission and neuroinflammation (Table 2).

Table 3 shows the percent of change by VPA compared to controls for each gene in males, whether the change was corrected by SAM administration or not. VPA changed the expression of 3 genes, 2 were upregulated and one was downregulated. The biological pathway of these genes varies, as seen in the table. None of these genes was reported in SFARI database for genes reported in human ASD.

### SAM Administration to Controls

SAM alone did not affect gene expression in males. In females, only one gene, Lsr (lipolysis stimulated lipoprotein receptor) was significantly downregulated by SAM administration (adjusted *p* value 0.008764).

SAM administration to VPA exposed mice: Co-administration of SAM with VPA normalized in females the expression of 8 of 15 genes that were significantly affected (for at least 50%) by VPA and their expression was similar to their expression in controls (Table 2). Three of these genes were upregulated by VPA and 5 were downregulated. All 4 genes that were reported in the Sfari database of human and animal genes involved in ASD were normalized by SAM.

In males, SAM corrected the expression of all three genes that were changed by VPA at least 50% (Table 3). Note that the gene Nts was similarly downregulated by VPA and corrected by SAM in both genders.

Table 4 shows 8 genes that were significantly changed in the same direction by VPA in both genders and the percent of change was similar. Of these genes, only Nts gene changed by more than 50% (Table 4). Two of these 8 genes were reported in the Sfari database of human and animal genes involved in ASD: Unc13a and Cacna1a. These two genes were normalized by SAM administration only in males.

## 3. Discussion

In this study we explored the regulation of gene expression in the prefrontal cortex of adult male and female mice by VPA and the possible correction of these changes by SAM. We found that early postnatal injection of VPA induced statistically significant differences (*p* < 0.05) in the expression of many genes. In females, the expression of 146 genes was changed, 15 of them being changed by more than 50%. In males, the expression of 19 genes was significantly changed by VPA, 3 of them being changed more than 50%.

VPA is a broad-spectrum antiepileptic drug with a wide range of clinical effects. While normally VPA is considered to be a safe drug with only few side effects, during pregnancy it may cause severe teratogenic effects in the embryo and fetus [11]. VPA is a strong HDAC inhibitor, playing an important role in transcriptional regulation; thus it is reasonable to assume that gene expression changes detected in the prefrontal cortex of 60 day old adult mice results from VPA induced epigenetic modifications accumulating during early postnatal developmental stages in the brain. Indeed VPA –induced changes in gene expression were reported by several investigators [31,32,33].

S-adenosyl methionine (SAM) is an important biological methyl donor which provides the methyl groups for histone or nucleic acid modification and phosphatidylcholine production, tuning regulation of gene expression [51].

In the current study we treated VPA injected pups with SAM. In males, 75% of VPA induced differently expressed genes were “repaired” by SAM administration; in females more genes were affected by VPA and only 52% of VPA- affected genes normalized by SAM. These results correlate with our previously reported behavioral findings that demonstrated greater beneficial effect of SAM in males in most of the behavioral tests compared to females [23]. In the behavioral analysis of our study, a composite ASD score combined of the social novelty preference, T-maze test and self-grooming assay was calculated for each mouse. An elevated score in the VPA group with correction in the SAM treatment group was noted both in females and males. However, differences were higher in males compared to females.

Male-to-female prevalence ratio of ASD is estimated to be about 4.5 [52]. This strong male predominance suggests the existence of different genetic involvement in the pathogenesis of ASD. In our study, the VPA –induced changes in gene expression were indeed different between the genders.

Most studies using the VPA model have predominantly explored the male sex and gender differences in autistic like behavioral phenotypes were reported only in few studies. Autistic male mice and rats were reported to display less social interest and had impairment in social novelty preference [28,53], decreased ability of nest-building [54] increased anxiety like behaviors, increased repetitive/stereotypic-like activity and lower sensitivity to pain [24]. Only few studies reported impaired social behavior in females as that was generally less pronounced than in males [55].

In our previous study, both genders exhibited autistic-like behaviors with differences in several neurological functions. Females were more prone to present anxiety-related traits as observed in the open field (center duration) and Elevated plus maze (head dipping); males displayed greater impairment in social novelty preference, grooming frequency and cognitive rigidity (T-maze) implying that the autistic-like behavior was stronger in males in spite of fewer VPA-affected genes. The differences were also reflected in the higher autism composite score induced by VPA in males [23].

We will therefore discuss the role of the genes whose expression was significantly changed by VPA and corrected by SAM and are already known to be related to ASD and/or to several common psychiatric and neurodegenerative diseases of the brain.

Only one gene, Nts, was similarly downregulated by more than 50% in males and females and its expression was normalized by SAM administration. Hence, this gene deserves special consideration.

Neurotensin (NTS) is a 13-amino-acid peptide which acts as a principal neuromodulator in the central nervous system. NTS was reported to be involved in mediating visceral analgesia [56], vocal communication and social behaviors [57], maternal aggression during offspring defense [58], sleep-wake regulation, anxiety, depressive-like behaviors [59] and locomotor activity [60]. The Nts system has been implicated in the pathophysiology of several psychiatric and neurological disorders, such as schizophrenia [61], anxiety [62] and depression [63]. Nts-expressing neurons modulate and interact with major neurotransmitter systems, including the dopaminergic [64,65], glutamatergic [66,67,68], gabaergic [69] and cholinergic [70] systems. NTS has strong interactions with dopamine and in the rat prefrontal cortex Nts is exclusively localized in dopamine axons [71].

In mammals, the Nts biological effects are mediated through two G-protein coupled receptors, Nts1 and Nts2, and one single-transmembrane receptor, Ntsr3 /sortilin [61,72,73]. Activation of Nts1 receptor by the selective agonist PD149163 reduced locomotor activity in mice with dose dependent effects [74]. This agonist also improved memory performance in male Norway rats [75]. Two studies reported elevated NTS levels in the serum of autistic children [76,77] that may occur as a result of inflammatory processes [78,79].

Although Nts seems to be involved in pathways and brain areas critical for social behavior, it received very little research attention regarding its possible role in the etiology of ASD. Kirsten et al. established a mouse model for ASD by prenatal Lipopolysaccharide exposure and reported autistic like behavior and reduced NTS (protein) plasma levels in male offspring. Postnatal treatment with the anti-diabetic drug Pioglitazone corrected social and communication deficits and abolished the reduction in the NTS levels [80,81]. We found reduced Nts mRNA levels in VPA exposed male and female mice that were corrected by SAM administration. In our previous behavioral analysis we detected increased anxiety-like behaviors, manifested by reduced frequency of head dipping in the VPA exposed male and female mice in the elevated plus maze test. A possible connection between reduced NTS protein activity and lower performance in the same behavioral test was described also by Normandeau et al. [82] in a mouse model of chronic stress.

In our previous study [23] we also detected significantly decreased time spent in the center of the arena in the open field, especially in females. These anxiety-related behaviors normalized by co-administration of SAM. Similar anxiety-related phenotype in the open field activity test was described by Fitzpatrick et al. [59] in Nts Receptor1 knockout male mice, with less distance traveled in the open field, less time spent in the center and more time spent in the corners than the wild-type controls.

Since Nts was the single gene similarly downregulated by VPA more than 50% and corrected by SAM in both genders, it is tempting to presume that downregulation of Nts may induce autistic like behavior. As Nts was not yet reported in Sfari data base as possibly related to ASD in animals or man, this issue needs further studies.

In addition to Nts, we have detected 8 genes that changed in both genders, with significant statistical difference from control (but less than 50% change). The similarity in the VPA-induced changes and the fact that both genders had ASD like behavior, may imply that several of these genes are indeed related to the autistic like behavior of the VPA treated animals. Two of these genes: Unc13a and Cacna1a also appear in Sfari database for genes reported in human/animal ASD. Since SAM improved the autistic –like behavior in both genders, we may decrease the number of candidate genes only to those that were corrected by SAM.

The expression of Unc13a and Cacna1a was upregulated in both genders, but normalized by SAM only in males. Unc13a, a neurotransmitter release regulator at nerve cell synapses, single-nucleotide exchange in the Unc13a gene, was reported in a number of disorders including delayed cognitive development, speech impairment, ASD, and ADHD [83].

Cacna1a encodes for the alpha-1A subunit of a neuronal ion calcium channel, which is predominantly expressed in neuronal tissue. CACNA1A loss-of-function mutations are associated with cognitive impairment including intellectual disability, ADHD and ASD [84]. Both genes are involved in glutamatergic synaptic neurotransmission and significantly associated with major depressive disorder [85]. Glutamate is considered to be a central excitatory neurotransmitter in CNS synaptic transmission and is involved in learning, memory and synaptic plasticity. Hence, the gender differences in the gene expression involved in glutamatergic pathways may explain the sex differences in the pathology of ASD.

Adults with ASD have regional abnormalities in subcortical glutamatergic neurotransmission that are associated with variations in social development [86]. Reduction in glutamate metabolism in ASD was reported in the basal ganglia [86], anterior cingulate cortex and the thalamus [87], while increased glutamate metabolism was found in the amygdala-hippocampal region of ASD individuals [88].

In a Shank2-mutant mouse model of ASD, Won et al. reported marked decrease in NMDA (*N*-methyl-*D*-aspartate) glutamate receptor (NMDAR) function which was accompanied by ASD-like behaviors including impaired social interaction, reduced social communication by ultrasonic vocalizations, and repetitive behavior, increased anxiety-like behavior and impaired spatial learning and memory. Furthermore, treatment of mice with a positive allosteric modulator of metabotropic glutamate receptor 5 (mGluR5) which normalized NMDAR function were found to have enhances social interaction.

A recently published review by Wickens et al. [89] described sex differences in the glutamatergic system contribution to psychiatric diseases with male predominance in schizophrenia, ASD and ADHD, and female predominance in Alzheimer’s disease and major depressive disorder. Furthermore, they described overall increase in glutamatergic transmission in females in most of these diseases that may play a protective role and lead to differences in symptomatology. Our findings support the involvement of sex specific alternations in the glutamatergic system in ASD pathophysiology. More studies are needed to estimate how glutamate dysfunction differentially affects males and females in context to ASD phenotype.

### 3.1. Genes Whose Expression Was Significantly Changed in Females

We will focus on genes previously reported in ASD: Drd1, Adora2a, Grin2a and Flt1. The expression of Drd1 and Adora2a was decreased and the expression of Flt1 and Grin2a was increased after VPA injection. All these genes were normalized by SAM administration. In addition, we will also focus on Nlrp3 gene that has been upregulated by almost 250% from controls and on Chat gene- of importance in acetyl choline metabolism that was downregulated by 55% and normalized by SAM.

The dopamine D1 receptor (Drd1) is among the most important postsynaptic effectors of dopamine function in the central nervous system. Drd1 regulates neuronal growth and development, mediates behavioral responses, and is involved in modulation of Drd2 mediated events. Drd1 was reported to be involved in social cognition [90]. Mutant Drd1 male rats have significantly reduced sociability and decreased interest in social novelty accompanied by decreased ultrasonic vocalization [91].

Adora2a: Adenosine A2A receptors are closely intertwined with the dopamine neurotransmitter system, are co-localized and have functional interactions with dopamine D2 receptors [92,93].

Adora2a is located on 22q11.23 chromosome, while deletions and duplications of chromosome 22q11.2 are associated with higher rates of ASD and psychotic symptoms [94]. Polymorphisms in Adora2a have been associated with schizophrenia, psychosis and anxiety [95,96,97]. Furthermore, Freitag et al. [98] found an association between single variant of Adora2a gene and increased autistic symptoms in human. Squillace et al. [99] revealed dramatically blunted Drd2 and Adora2a neurotransmission in BTBR strain of autistic mice, a finding that could play a role in the social deficits exhibited by these animals.

The Grin2a gene encodes the glutamate-binding GluN2A subunit of the N-methyl-D-aspartate receptors (NMDARs)- a glutamate gated cation channels that mediates the slow component of excitatory synaptic transmission [100]. NMDARs play essential roles in normal brain function, including learning, memory, synaptic plasticity, motor and sensory processes, and nervous system development. Grin2a mutations were identified in children with specific language impairment, speech disorders and epilepsy [101,102]. In our study VPA induced upregulation of Grin2a gene. Up regulation of GRIN2A receptor was also found in the prefrontal cortex of patients with major depression [103,104].

Flt1 (VegfR1) - Vascular endothelial growth factor receptor- 1 is a Tyrosine-protein kinase that acts as a cell-surface receptor for VEGF, a well-known major angiogenesis stimulating factor. Flt1 was reported to be involved in the regulation of angiogenesis, cell survival and cell migration, and acts as a negative regulator of embryonic angiogenesis [105,106]. In our study, the expression of Flt1 gene was upregulated by 83% after VPA exposure, and normalized by SAM. Higher prefrontal cortex expression of Flt1 was associated with worse cognitive trajectories in Alzheimer disease patients [107].

Upregulation of the Flt1 gene was also described by Hu et al. [108] in a study that examined the gene expression profiling on DNA microarrays of lymphoblastoid cell lines derived from five monozygotic twin pairs discordant in severity for ASD and language impairment. There was a positive association between the severity of the autistic phenotype exhibited by the twins (higher or lower expression in the more severely affected twin relative to the other twin) and the expression level of several genes involved in neurological function, including Flt1.

The expression of Chat was reduced by 56% in females but not in males. However, we decided to focus on this gene because of its importance in neuronal pathways.

Choline acetyltransferase (Chat) encodes CHAT enzyme which catalyzes the biosynthesis of the neurotransmitter acetylcholine. Acetylcholine is one of the most important neurotransmitters controlling the parasympathetic and the sympathetic autonomic nervous system as well as the somatic motor system. Furthermore, Acetylcholine is a major neurotransmitter in the CNS, involved in many functions, including control of locomotor activity, emotional behavior, and higher cognitive processes such as attention, learning and memory processes [109,110,111]. Changes in cholinergic neurotransmission are associated with a variety of important neurological disorders including Alzheimer’s disease, schizophrenia, Parkinson’s disease, epilepsy, attention-deficit hyperactivity disorder, mild cognitive impairment and also ASD [110,112]. Cholinergic agents such as Donepezil have been proposed for treatment of ASD symptoms [113].

Wang et al. [114] examined the role of the nicotinic cholinergic system on social and repetitive behavior in BTBR mouse model of ASD. They treated autistic mice for 4 weeks with different doses of nicotine (50, 100, 200 and 400 μg/mL), an agonist of nicotinic acetylcholine receptor subtypes that is also known to upregulate the expression of various nAChR (acetylcholine receptor) subtypes. They reported that high doses of nicotine (200 and 400 μg/mL) significantly decreased repetitive self-grooming behavior in BTBR mice compared to baseline, while lower doses of nicotine (100 μg/mL) increased social interactions in BTBR mice in the three chambers social interaction test. The authors combined male and female data because they did not find any gender differences. In our previous study males, but not females, exposed to VPA showed significantly increased grooming frequency and lower preference for social novelty and preference to familiar social stimuli in the Social interaction test, which was normalized by the co-administration of SAM [23].

Although, dysfunction of the cholinergic system may underlie autism-related behavioral symptoms, we did not find any evidence from the literature connecting between the involvements of Chat to ASD etiology, especially not in VPA-induced ASD. Furthermore, studies analyzing tissue samples from deceased autistic adults reported no alteration of CHAT biochemical activity in the cerebellum [115] and in the frontal and parietal cerebral cortex [116]. Despite these reports, studies of other neurological disorders suggested significant involvement of CHAT enzyme alternations and described its possible connection to behavioral changes. Hence, further studies should be carried out to elucidate the possible role of Chat in the etiology of ASD.

Down-regulation of Chat as well as other cholinergic signaling genes was demonstrated in a chronic restraint stress rat model of depression, in which rats display depression-like behaviors such as anhedonia and mood despair [117]. In Alzheimer patients Chat activity is significantly decreased in the cerebral cortex and hippocampus and it seems to correlate with the severity of the dementia [118].

In our previous study [23], in the water T maze assessing reversal learning, cognitive rigidity and repetitive behavior, the learning curve for detecting the new location of a hidden platform was significantly less effective and there were greater latencies in trials only in the VPA-exposed females compared to controls. This higher latency was normalized by the co-addition of SAM. In males we did not find any differences in the water T maze test, in parallel to normal expression of Chat gene.

### 3.2. Genes Whose Expression Was Significantly Changed Only in Males

In addition to Nts 3, Ryr1 and Itga7 were all upregulated by VPA. The expression of all these genes was normalized by SAM administration. None of these genes was reported to be related to ASD.

Ryr1- ryanodine receptor 1 plays a central role in the regulation of intracellular calcium (Ca^2+^) homeostasis, which is crucial for neuron survival and function. Ryrs exist in three isoforms (Ryr 1–3). Ryr2–3 is predominantly expressed in Purkinje cells of the cerebellum and cerebral cortex and in the hippocampus [119] and is involved in modulation of learning and memory functions [120]. Ryr1 also plays a critical role in calcium release and muscle contraction in skeletal muscle [121].

Gene ontology functional enrichment analysis for the genes significantly changed by VPA in females, demonstrated potential shared mechanism with neurodegenerative diseases such as Alzheimer’s disease and Huntington’s disease, as our Nlrp3 analysis suggest. Thus, GO term analysis suggests that common pathways underlay common symptoms in different pathologies. Studying such common pathways may have implication to several diseases including ASD. Nevertheless, we could not find shared pathways between females and males. However, it is still possible that individual genes whose expression was similarly changed by VPA and corrected by SAM play a role in the autistic like behavior of both genders. Such a candidate gene is Nts. The differences in the behavioral pattern between VPA treated males and females may be related to the change in the expression of genes that are differently affected between genders or not corrected by SAM. However, from our data it is impossible to point out to the genes that are specifically responsible for the ASD like behavioral changes.

## 4. Materials and Methods

### 4.1. Animals

Male and female (four days old) outbred ICR albino pups were injected subcutaneously on postnatal day 4, either with 300 mg/Kg body weight of VPA dissolved in normal saline or normal saline (NS). The dose chosen was the minimal dose that was reported to produce ASD in mice offspring. This time is developmentally equivalent to months 7–8 of human pregnancy. Each of these two groups (VPA and NS treated) were further subdivided into two groups- one receiving daily by intraoral gavage, from day 5 for 3 days, 30 mg/Kg body weight of SAM dissolved in NS, and the other receiving NS. Each treatment or control group consisted of 12–16 males and a similar number of female mice.

Twenty four hours after VPA injection pups were given daily for 3 consecutive days 30 mg/kg SAM by intragastric lavage. On day 60, the animals were euthanized; brains (prefrontal cerebral cortex) were removed for molecular studies. All pups were handled similarly in the animal quarters under optimal temperature and light. The University of Ariel Ethics committee for experiments on animals received approval for the study (IL-109-06-16, 6th July 2016). The more complete data on the experimental design and the results of the neurobehavioral tests were published elsewhere [23].

### 4.2. RNA Extraction and Gene Expression Analysis

Total RNA was extracted from the right prefrontal cortex of the mice using the RNA/DNA/protein purification plus kit (47700; Norgen, Thorold, ON, Canada) according to the manufacturer’s protocol. RNA was quantified at absorbance of 260nm. An OD260/280 ratio between 1.8 and 2.2 was considered for further processing.

Gene expression analysis was performed for 24 samples of each gender, 6 in each group, using the NanoString nCounter system (NanoString Technologies, Seattle, WA, USA) that provides a simple way to profile specific nucleic acid molecules in a complex mixture. The system is based on direct digital detection of mRNA molecules utilizing target-specific, color-coded probe pairs that can hybridize directly to target molecules. The expression level of mRNA molecules is measured by counting the number of times the barcode for that molecule is detected by a digital analyzer. It does not require the conversion of mRNA to cDNA by reverse transcription or the amplification of the resulting cDNA by PCR. The system does not need amplification and is sensitive enough to detect low abundance molecules.

The data is expressed by the number of mRNA molecules In 100 ng/uL of RNA. It can simultaneously quantify up to 800 different interesting targets in a single reaction, making it ideal for miRNA profiling and targeted mRNA expression analysis [122]. We used the Mouse Neuropathology Panel that includes 770 genes covering pathways involved in neurophysiology, neurodegeneration and other nervous system diseases, and 10 internal reference genes for data normalization.

### 4.3. Gene Functional Enrichment

Gene Ontology for Functional enrichment of pathways (KEGG) was performed for genes which were found to be significantly altered by VPA using DAVID bioinformatics resources 6.8 [123]. The total list of 770 genes related for neuropathology that were tested in the array was used as background. Significantly enriched pathways were selected by *p* value (<0.05) and fold enrichment (>1.3) [124,125].

### 4.4. Statistical Analysis

NanoString analysis was performed on 6 samples from each group and each gender. This large sample size allows detecting and referring also to small but statistically significant changes in gene expression.

Gene expression data were analyzed by two-way ANOVA, in the first time with control as the reference group, and the second time with VPA as the reference group.

NanoString analysis was performed on 6 samples from each group and each gender. This large sample size allows detecting and referring also to small but statistically significant changes in gene expression.

Gene expression data were analyzed by the R package DESeq2, v1.22.1 [126]. Since samples were measured in two batches, the statistical model included both the treatment and the batch. After normalization by the internal reference genes, Wald test was used to compare the different conditions, using default parameters, including the significance threshold of Benjamini-Hochberg FDR (p adj) less than 0.05. Further filtering of significant genes required a change in expression of at least 50% relative to the control group.

NanoString assay is reported to be reliable, high-throughput assay used to simultaneously screen for gene expression changes in clinical practice [127,128]. Studies that used the NanoString n-counter system usually did not use rtPCR analysis for validation [129,130,131,132].

## 5. Conclusions

Most of the genes whose expression were changed by VPA and corrected by SAM seem to be involved in ASD and/or several other diseases of the nervous system, or in cognition and memory. These include genes involved in neuronal function and in inflammation. Genes of these two groups were also described as being associated with human ASD. The fact that SAM normalized their function may also explain the reversal of the autistic like behavior and reduction of brain oxidative stress observed in our previous studies. The changes induced by VPA in several pathways (Alzheimer’s disease and Huntington’s disease) were observed only in females. Since these changes are gender specific, they might not be related to the ASD like behavioral changes manifested in females and males. This emphasizes that only changes in the expression of individual genes may be related to the autistic like behavior. Unfortunately, our data do not enable us to correctly point to the gene/s that may be responsible for the autistic like behavior induced by VPA, but point to a number of genes possibly involved, especially those similarly changed by VPA in both genders and “corrected” by SAM such as Nts or Myrf. Early postnatal administration of SAM alone had very little effect on gene expression. Methylation of DNA occurs via the methyl donor SAM, and maintenance of adequate SAM circulating concentrations are, in part, dependent on folic acid and vitamin B12 [133]. Because most of the epigenetic programming accrue during prenatal development, it is reasonable that SAM effects as a methyl donor on the methylation pattern will be lower after birth. Indeed, in a different study we found that SAM administration during mid-pregnancy to ICR mice caused many significant changes in gene expression in the prefrontal cortex of the neonates (to be published).

## Figures and Tables

**Figure 1 ijms-20-05278-f001:**
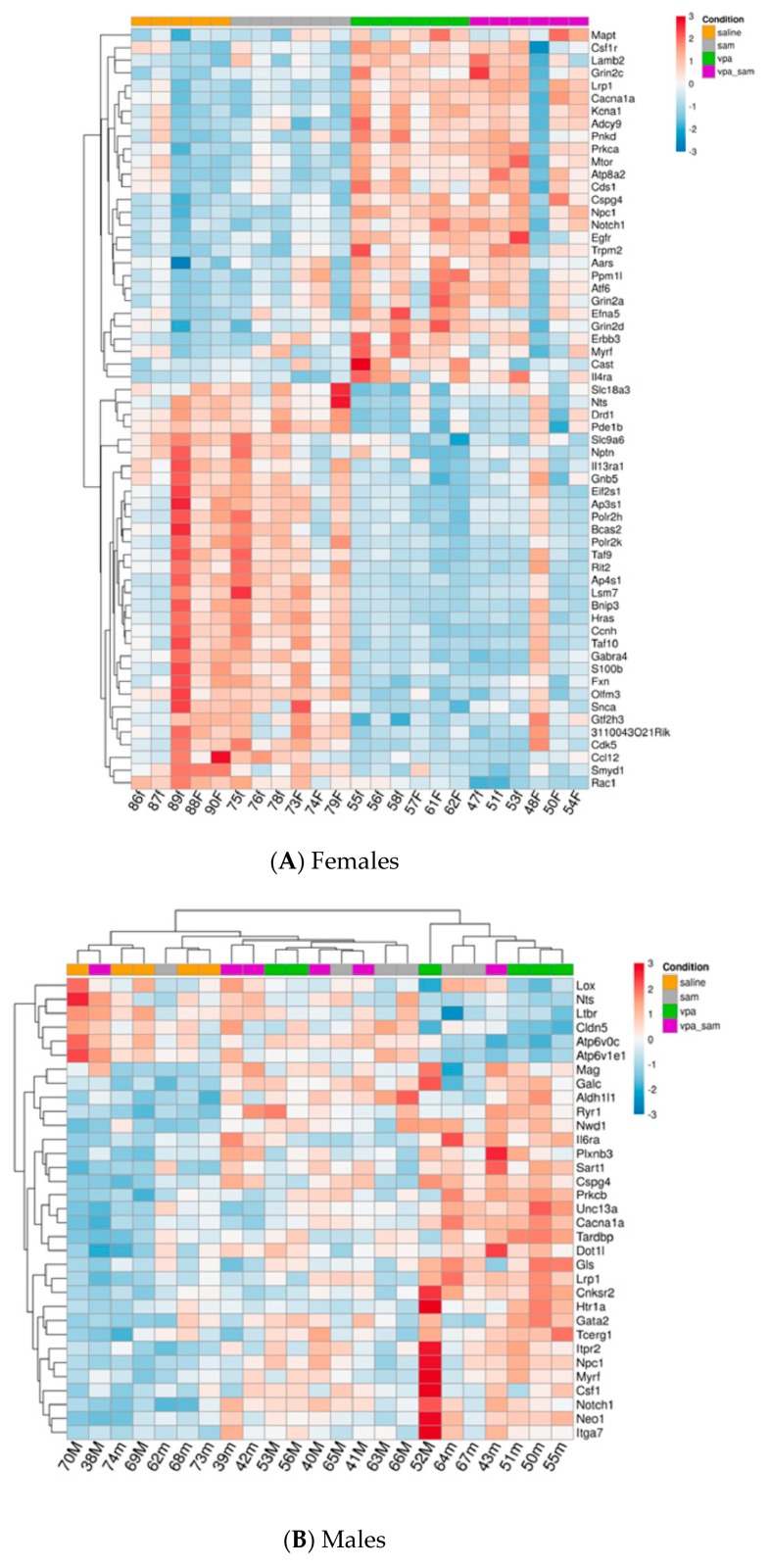
Effect of VPA and SAM administration on the gene expression in the prefrontal cortex. Heat map of the genes significantly changed by VPA in the different groups: **A**. females, **B**. males. Each heat map consists of two NanoString panels (12 samples).

**Figure 2 ijms-20-05278-f002:**
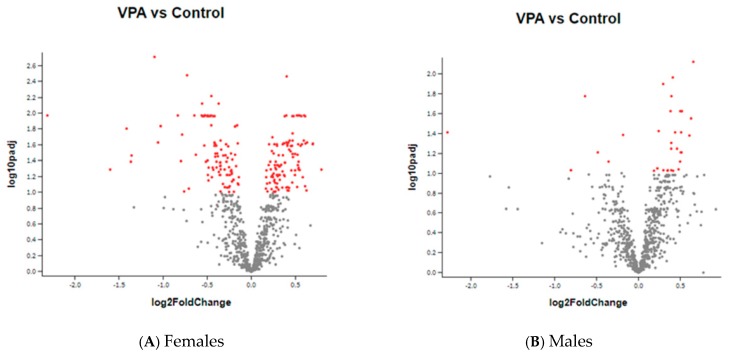
(**A**,**B**) are a volcano plot of log 2 fold-change (x-axis) versus −log 10 adjusted *p*-value (y-axis, representing the probability that the gene is differentially expressed). Every dot represents one gene. Red dots: genes with statistically significant change compared to controls. Grey dots: no statistical change. More genes were up or down regulated in females compared to males. In males, most of the genes whose expression was changed by VPA were upregulated.

**Table 1 ijms-20-05278-t001:** Functional enrichment analysis of the genes significantly changed by VPA in females and males.

	Category	Term	Count	%	*p* Value	Genes	Fold Enrichment
	KEGG_PATHWAY	mmu04510: Focal adhesion	22	6%	0.011	PRKCA, EGFR, COL4A2, HRAS, COL4A1, FLT1, MAP2K1, PIK3CB, IGF1, PRKCG, BAD, CTNNB1, IGF1R, MAPK1, LAMB2, TNR, ITGA7, RAC1, PDGFRB, PIK3CA, PAK1, FN1	1.630
	KEGG_PATHWAY	mmu03022: Basal transcription factors	9	3%	0.025	TAF10, MNAT1, CCNH, GTF2H3, TAF9, CDK7, TAF6L, GTF2B, TBPL1	2.222
	KEGG_PATHWAY	mmu05016: Huntington’s disease	18	5%	0.025	POLR2H, HTT, CREBBP, DNAH1, SOD1, PPARGC1A, TFAM, AP2B1, EP300, PLCB4, HDAC2, GNAQ, CASP9, SP1, GRIN2B, BAX, TBPL1, HAP1	1.626
**Females**	KEGG_PATHWAY	mmu05215: Prostate cancer	17	5%	0.026	EGFR, HRAS, MAP2K1, PIK3CB, CREBBP, IGF1, BAD, CTNNB1, IGF1R, MAPK1, CDKN1A, ATF4, EP300, CASP9, PDGFRB, PIK3CA, MTOR	1.657
	KEGG_PATHWAY	mmu04151: PI3K-Akt signaling pathway	31	9%	0.029	HRAS, IL4RA, PPP2R5C, IGF1R, LAMB2, CASP9, TNR, RAC1, PIK3CA, PRKAA2, INSR, CSF1R, FN1, PRKCA, EGFR, COL4A2, FLT1, COL4A1, MAP2K1, PIK3CB, IGF1, BAD, MAPK1, CDKN1A, ATF4, ITGA7, PDGFRB, GNB5, EFNA5, PPP2R5E, MTOR	1.383
	KEGG_PATHWAY	mmu05010: Alzheimer’s disease	21	6%	0.038	CDK5R1, ADAM10, SNCA, GRIN2A, BAD, CDK5, ATF6, MAPK1, APP, PLCB4, LRP1, GNAQ, CASP9, GRIN2B, GRIN2C, MAPT, RYR3, GRIN2D, BACE1, PPP3CC, CACNA1C	1.496
	KEGG_PATHWAY	mmu04020: Calcium signaling pathway	22	6%	0.044	PRKCA, EGFR, SLC8A1, DRD1, ADORA2A, ERBB3, GRIN2A, PRKCG, ATP2B3, PLCB4, GNAQ, ADCY9, PDE1B, GRIN2C, RYR3, GRIN2D, PDGFRB, PPP3CC, RYR2, CACNA1C, CACNA1A, CACNA1B	1.455

**Table 2 ijms-20-05278-t002:** Genes changed by VPA at least by 50% in females and the normalization by SAM *.

Gene.	Official Full Name	Up (+)/down(−) Regulated by VPA	% Change	Adjusted *p* Value	Neuroinflammation	Neuroplasticity, Development & Aging	Metabolism	Compartmentalization and Structural Integrity	Neurotransmission		Normalized by SAM Administration
Smyd1	SET and MYND domain containing 1	(−)	−63%	0.001951	−	+	−	−	−		−
Cspg4	chondroitin sulfate proteoglycan 4	(+)	51%	0.01074	+	+	−	−	−		−
Nts	neurotensin	(−)	−71%	0.01074	−	−	−	+	−		+
Il4ra	interleukin 4 receptor, alpha	(+)	63%	0.01097	+	−	−	−	−		−
Ccl12	chemokine (C-C motif) ligand 12	(−)	−58%	0.01461	+	+	+	+	+		−
Drd1	dopamine receptor D1	(−)	−57%	0.01567	−	−	−	+	+	Reported in human and animal SFARI data base.	+
Slc18a3	solute carrier family 18 (vesicular monoamine), member 3	(−)	−58%	0.02347	−	−	−	+	+		+
Notch1	notch 1	(+)	51%	0.02426	−	+	−	−	−		−
Ppm1l	protein phosphatase 1 (formerly 2C)-like	(+)	60%	0.02426	+	−	−	−	−		+
Grin2a	glutamate receptor, ionotropic, NMDA2A (epsilon 1)	(+)	61%	0.02478	−	−	−	+	+	Reported in human SFARI data base.	+
Chat	choline acetyltransferase	(−)	−56%	0.03445	−	−	+	+	+		+
Cd40	CD40 antigen	(−)	−55%	0.04037	+	−	−	+	−		−
Cd4	CD4 antigen	(−)	−55%	0.0411	−	−	−	+	−		−
Adora2a	adenosine A2a receptor	(−)	−59%	0.05169	−	−	−	−	+	Reported in human and animal SFARI data base.	+
Flt1	FMS-like tyrosine kinase 1	(+)	83%	0.05169	+	+	−	−	−	Reported in human SFARI data base.	+

* The genes are ordered by the significance of the adjusted *p* value.

**Table 3 ijms-20-05278-t003:** Genes changed by VPA at least by 50% in males and the normalization by SAM *.

Gene.	Official Full Name	Up(+)/down(-) Regulated by VPA	% Change	Adjusted *p* Value	Neuroinflammation	Compartmentalization and Structural Integrity	Neurotransmission	Normalized by SAM Administration
Ryr1	ryanodine receptor 1, skeletal muscle	(+)	61%	0.007544	−	−	+	+
Nts **	neurotensin	(−)	−76%	0.03878	−	+	−	+
Itga7	integrin alpha 7	(+)	53%	0.04183	−	+	−	+

* The genes were ordered by the significance of the adjusted *p* value. ** Nts was also similarly downregulated in females.

**Table 4 ijms-20-05278-t004:** Genes whose expression was significantly changed in the same direction by VPA in males and females and their correction by SAM *.

Gene.	Official Full Name	% Change Males	Adjusted *p* Value- Males	% Change Females	Adjusted *p* Value- Females	Neuroplasticity, Development & Aging	Metabolism	Compartmentalization and Structural Integrity	Neuron-Glia interaction	Neurotransmission	Normalized by SAM Males	Normalized by SAM Females	
Npc1	Niemann-Pick type C1	31%	0.01089	26%	0.003442	−	+	−	−	−	+	−	
Plxnb3	plexin B3	34%	0.02376	40%	0.0259	+	−	−	−	−	−	+	
Unc13a	unc-13 homolog A (C. elegans)	0.24	0.02376	0.23	0.04218	−	−	+	−	+	+	−	Reported in human SFARI data base
Myrf	myelin regulatory factor	48%	0.0281	30%	0.02347	−	−	−	+	−	+	+	
Notch1	notch 1	0.43	0.03878	0.51	0.02426	+	−	−	−	−	−	−	
Nts	neurotensin	−76%	0.03878	−71%	0.01074	−	−	+	−	−	−−	+	
Itga7	integrin alpha 7	0.53	0.04183	0.3	0.04599	−	−	+	−	−	+	−	
Cacna1a	calcium channel, voltage-dependent, P/Q type, alpha 1A subunit	25%	0.0497	36%	0.02478	+	−	+	−	+	+	−	Reported in human SFARI data base

* The genes were ordered by the significance of the adjusted *p* value.

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
