# Peer review of "Gender Related Changes in Gene Expression Induced by Valproic Acid in A Mouse Model of Autism and the Correction by S-adenosyl Methionine. Does It Explain the Gender Differences in Autistic Like Behavior?"

_ijms, 2019, doi:10.3390/ijms20215278_

Round 1

Reviewer 1 Report

I think the manuscript includes new and intriguing contents, however the authors should revise it according to the following concerns;

1.The authors should discuss on the putative differences between ASD model induced by VPA and human (real-world) ASD if any.

2.As mentioned in the introduction, ASD is more prevalent in male with the average male to female ratio 4:1.

Can the present results explain this male bias? The authors should discuss more in detail on this point.

3.Why was the prefrontal cortex selected to analyze changes in the expression of genes induced by VPA?

Author Response

We would like to thank the reviewers for their important and enriching comments. We will address their comments one by one. All changes were left in color and the lines were cited.

Reviewer 1: 

We added in the introduction paragraph discussing the differences between ASD in human and the VPA model and gender differences (lines 65-74). Our data cannot explain the gender differences in the prevalence of ASD since more genes were changed by VPA in females compared to males while ASD is more prevalent in males. However, gender differences are smaller in the VPA induced ASD like behavior compared to human ASD. These points were discussed in the manuscript (lines 557-559; 575-579). We explained in the introduction why we chose the prefrontal cortex for our studies (lines 118-132).

Reviewer 2 Report

Dear editor

The manuscript by Liza et al reports a differential mRNA expression for a number of genes that are involved in the autism like behavior in a VPA induced mice model. 

The design of the study and the technical quality of the work appear convincing and results can be somehow of general interest. Authors used a good number of animals and analyzed data using correct statistical approaches.

However, there is a number of major points that affects the overall merit of this work to be at the level that can be accepted for publication at IJMS:

-My main concern is the poor conclusion that isn’t supported by the presented data. Data failed to correctly point out to the genes that may be related to the autistic like behavior induced by VPA. However, authors made a claim that the behavioral and other differences between genders may be related to genes that were differently affected by VPA in males and female and/or differently affected by SAM.

-Authors have reported a good number of genes but they haven't validated any of them using RT-PCR. Moreover, authors haven't validated the protein expression of any single protein. The discrepancies between gene and protein data is a well-known fact and since the aim of this study was to provide a mechanistic insight, authors need to investigate two or more proteins to solidify the outcomes of the suggested effect. Nts might be a good candidate as it was similarly down-regulated in both genders.

-The significance of this work would be significantly enhanced if the authors have performed GO and pathway analyses (e.g. KEGG) since this might point out to an interesting mechanistic insight.

-The discussion is quite long and it mainly consists of reporting a well-known function(s) of all the differentially expressed gene without providing a mechanistic insight and/or how they might be functionally linked together.

Best.

Author Response

We would like to thank the reviewers for their important and enriching comments. We will address their comments one by one. All changes were left in color and the lines were cited.

Reviewer 2:

This reviewer suggested to validate some of the VPA- induced changes in gene expression by rtPCR. This was not done by us because the method of Nanoastring N-counter is considered very accurate and most studies do not verify the results by rtPCR. We added a paragraph related to that issue and several references at the end of the methods section (lines 575-578). Unfortunately, we do not have more RNA or protein to repeat studies since our deep freezer (-80 C) broke and all the stored material is no more valid. As suggested by this reviewer, we now performed GO and pathway analyses but, after correction for multiple data using the Benjamini-Hochberg FDR, there were no differences, as stated in table 1 which is now added at the beginning of the results section. The method was also described in the methods section (lines 160-171 and 555-559). We agree with this reviewer that our results do not allow us to point out to the specific genes that might be responsible for the ASD like behavior. To the best of our knowledge, this problem is common to most studies describing the possible involvement of genes in animal models of ASD. This was stated by us at the end of the discussion (lines512-520) and in the conclusions. We decided to describe in the discussion some of the literature on those genes that are known to be related to ASD or to other psychiatric disorder and their expression was altered by VPA and corrected by SAM. It is not clear if this reviewer suggests to shorten the discussion. However, we made several changes in the discussion (i.e. lines 328-331 related to Nts) and removed several paragraphs and references. Unfortunately, as stressed by us in several places, our data do not enable us to specifically point to the genes responsible for the behavioral alterations in the treated animals, except Nts - a good candidate gene. We believe that the correction by SAM of the VPA – induced changes in gene expression is the important contribution of this study. We could show that the physiologic methyl donor may effectively reverse the effects of VPA on gene expression and consequently ameliorate the ASD like behavior in the treated animals.

We thank again the reviewers for their important comments.  

Round 2

Reviewer 1 Report

I think the manuscript has properly been revised according to the reviewer's comments.

Author Response

We would like to thank reviewer 1 for his approval of the revised version of the manuscript.   

Reviewer 2 Report

Dear editor

I would like to thank the authors on their efforts towards addressing my concerns and suggestions. I think that the newly added sections and the provided referneces helped towards the improvement of the current version compared to their earlier submission. However, there is still a room for improvement based on the following comments and suggestions:

I agree with the fact that NanoString nCounter is better than most of the currently available conventional gene expression analysis methodologies but my suggestion has been made based on the type of conclusion for this study. NanoString nCounter is still a hybridization-dependent molecular barcoding technology which needs validation when follow-up research relies on this gene(s) of interest being differentially expressed. However, based on the fact that authors don't have an access to more samples based on the mentioned unfortunate incident and that the authors have provided some insights on overall patterns using (GO and KEGG analyses), then I will consider this to be somehow enough for the purpose of this study. One of my major concerns is that the authors include genes, GO terms and/or KEGG pathways even when the p values didn't hit the set satisicical cut-off of ≤ 0.05. Authors need to remove the following data from their results: The last KEGG pathway in Table 1. The last 5 genes in Table 2. The last gene (Lox) in Table 3. I would like to thank the authors for providing Table 1. However, there is no need to provide Pop hits and Pop totals in the revised submission. Moreover, the major determinants for identifying significant KEGG pathways are the p value and fold enrichment. Bonferroni and Benjamini won't neccessarily provide much an insight regarding the significane of a suggetsed KEGG pathway based on the total number of significantly expressed hits, total number of genes involved in this pathway and background. Authors should provide KEGG maps for what they think (hypothesize) to be the most suggested pathway (or two). This data should be provided as a supplementary figure. These two references should provide an insight on how to present the newly requested  data: https://www.ncbi.nlm.nih.gov/pubmed/28715131 and https://www.ncbi.nlm.nih.gov/pubmed/29311824.  Authors could remove Bonferroni and Benjamini colums from revised Table 1. Authors should add a line to section 4.3 stating that p values less than 0.05 and fold enrichment values were used to identify significant KEGG pathwats citing the two above mentioned papers. Discussion need to be shorted in general. Authors need to discuss how GO terms and KEGG could help in identifying a link between the network of their identifed DEGs.

Best.

Author Response

We would like to thank reviewer 1 for his approval of the revised version of the manuscript.    We will address the important comments of the second reviewer, hopefully fully addressing all his concerns.  All new changes were left in color after we incorporated the previous changes.

Reviewer 2:

We are indeed sorry for not having any RNA or protein for additional studies. As suggested by this reviewer, we had set our P value to be 0.05 or less instead of less than 0.1. Because of that, we changed all tables accordingly removing genes with a P value between 0.05-0.1. We were told by our Bioinformatics department that due to our correction for multiple data using the Benjamini-Hochberg FDR, p can be set to less than 0.1, as indeed described in several publications. Indeed, by setting a significant P values up to 0.05, the results did not change much although fewer genes were significantly changed by VPA in both genders. We made the requested changes in the KEGG pathway and therefore changed table 1 as suggested by this reviewer and added a supplementary figure 1 showing changes in 2 pathways – Huntington diseases and Alzheimer disease. The changes were observed only in females, implying that they are apparently not related to the ASD like behavior but to general pathological processes in the brain. We added the two suggested references. We shortened the discussion by removing several sentences and paragraphs (already incorporated in the text) and removing some of the discussion on Chat gene and the entire discussion on Nlrp3. These are marked in the text. The number of references was accordingly reduced. We thank again this reviewer and hope that we answered adequately all his concerns.

Round 3

Reviewer 2 Report

Dear Editor

I would like to thank the authors for critically addressing the raised concerns and suggestions.

I think that authors’ amendments made the revised version of this study acceptable for publication at IJMS.

Best